# The Lift Effects of Chordwise Wing Deformation and Body Angle on Low-Speed Flying Butterflies

**DOI:** 10.3390/biomimetics8030287

**Published:** 2023-07-03

**Authors:** Yan-Hung Fang, Chia-Hung Tang, You-Jun Lin, Szu-I Yeh, Jing-Tang Yang

**Affiliations:** 1Department of Mechanical Engineering, National Taiwan University, Taipei 10617, Taiwan; d10522021@ntu.edu.tw (Y.-H.F.); r10522112@ntu.edu.tw (C.-H.T.); f05525016@ntu.edu.tw (Y.-J.L.); 2Department of Aeronautics and Astronautics, National Cheng Kung University, Tainan 701401, Taiwan

**Keywords:** butterfly, lift, body angle, chordwise wing deformation

## Abstract

This work investigates the effects of body angle and wing deformation on the lift of free-flying butterflies. The flight kinematics were recorded using three high-speed cameras, and particle-image velocimetry (PIV) was used to analyze the transient flow field around the butterfly. Parametric studies via numerical simulations were also conducted to examine the force generation of the wing by fixing different body angles and amplifying the chordwise deformation. The results show that appropriately amplifying chordwise deformation enhances wing performance due to an increase in the strength of the vortex and a more stabilized attached vortex. The wing undergoes a significant chordwise deformation, which can generate a larger lift coefficient than that with a higher body angle, resulting in a 14% increase compared to a lower chordwise deformation and body angle. This effect is due to the leading-edge vortex attached to the curved wing, which alters the force from horizontal to vertical. It, therefore, produces more efficient lift during flight. These findings reveal that the chordwise deformation of the wing and the body angle could increase the lift of the butterfly. This work was inspired by real butterfly flight, and the results could provide valuable knowledge about lift generation for designing microaerial vehicles.

## 1. Introduction

The flight ability of insects is generally powered by them flapping their wings periodically. This completely differs from aircraft flight, which is powered by an engine with fixed wings [1,2,3]. Insects invariably maneuver their wings to fly swiftly, for example, to achieve immediate vertical flight, and to spontaneously alter their flight direction and hover [4,5,6,7,8,9]. For scientists and engineers to innovate a design for micro aerial vehicles (MAVs) [10,11,12], the amazing flying techniques of insects provide valuable ideas, such as flight at a small Reynolds number, small body size, and low flapping frequency. Thus, many scientists have been attracted to study the biomimetics of flight behavior and to obtain knowledge of the aerodynamic mechanisms of insects. The butterfly has two characteristics, i.e., two pairs of broad wings and a low flapping frequency, that differ from those of other insects, such as the bumblebee, mosquito, dragonfly, and cicada [13,14,15,16]. The wings of the butterfly are generally larger than those of other insects. The butterfly has two wings on each side of its body, the fore and hind wings, which partially overlap during flight. The flapping frequency of the butterfly is generally approximately 10 Hz, which is probably the lowest flapping frequency among insects [17,18]. The advantage of the pair of broad wings combined with the relatively low flapping frequency of the butterfly strongly increases the aerodynamic force and produces a powerful torque during a flap, which allows the butterfly to spontaneously generate a huge upward lift and downward pitch and to instantly alter the flight direction within a few flapping cycles [4,7,8].

Unlike other insects, the oscillation frequency of the butterfly’s abdomen is similar to its wing-beat frequency, which induces its body angle to control its flight dynamics [19]. These findings correspond well with the simulation studies that indicate that the body angle of the butterfly can effectively control its flight trajectory by altering the stroke plane angle and the direction of aerodynamic forces [11]. From the aerodynamic view, the detached leading- and trailing-edge vortices (LEVs and TEVs) merge in the wake region and form a three-dimensional vortex ring [19,20]. The vortex ring rotates toward the center, leading to a strong jet flow inside the vortex ring and propelling the butterfly’s flight. The direction of the jet flow can be altered by adjusting the body angle, thereby guiding the butterfly’s flight trajectory [11].

Many studies have indicated that the body angle of the butterfly becomes larger during the upstroke, which enhances the backward jet flow and creates stronger propulsion. These studies also illustrate that the maximum body angle of the butterfly during the upstroke lies between 30° and 50°, which theoretically brings the jet flow backward and upward [17,19,21,22]. Previous studies have also asserted that it is not only the body angle but also the rigid wing and body motion that influence the aerodynamics during butterfly flight [17,23,24]. In addition to these, the flexibility of wing deformation may also play an important role in the aerodynamics of butterfly flight [25,26,27].

Mountcastle and Daniel [25] employed a self-made flapping machine to examine the aerodynamics under varied flexibility of the butterfly with either a dry or wet wing. Their experimental results showed that the wet wing produced a larger added flow and, thus, increased the lift of the hawkmoth as compared with that of the dry wing. The wing deformation consists of twist and camber, which are deformations in the spanwise and chordwise directions, respectively. Zheng et al. [26] carried out experiments on butterfly (Vanessa cardui) flight in a wind tunnel to compare the aerodynamics and efficiency of different types of wings, including fully deformable wings, twist-only wings, and other flat-plane wings. Their simulation results showed that the twist-only wing had the best aerodynamics and the highest efficiency compared with the other wing deformation models during butterfly forward flight. The different simulated results in Suzuki et al. [27] indicated that an appropriate range of twisting stiffness of the spring produced a better average lift-to-drag ratio than rigid wings, which was due to the effect of chordwise deformation on flight dynamics. Such a result was also observed in an abiotic experiment conducted by Gehrke et al. [28]; they used membrane material to study the aerodynamics of a flapping wing. Their results showed that chordwise deformation enhanced lift during the entire flying cycle and that camber displacement between 15% and 20% had the optimal aeroelastic numbers. The importance of twist and camber deformation in influencing the aerodynamic effects of different insects has also been mentioned in many other studies [29,30,31,32,33,34].

The current knowledge of the aerodynamics of chordwise deformation and the effects of body angles during butterfly flight is still limited. To better understand the aerodynamics of chordwise deformation, the present work carries out a series of experiments on real butterflies to enrich the observation data of butterfly flight. In addition, a numerical simulation is conducted to examine the aerodynamics of a cambered wing at different body angles during butterfly flight in the hope that more valuable knowledge can be gathered for the design of MAVs.

## 2. Materials and Methods

### 2.1. Experimental Setup

A series of experiments, similar to those in the literature report [29], were carried out to record butterfly flight. Four blue tiger butterflies (*Tirumala septentrionis*) were placed inside a transparent acrylic box with dimensions of 100 cm (L) × 50 cm (W) × 60 cm (H), as shown in Figure 1a. Three high-speed cameras (Phantom v7.3, Phantom v310 and Ditect HAS-D73) were installed to record the butterflies’ flight. The speed of the three high-speed cameras was 1000 frames per second with a resolution of 1024 × 768 pixels, which enabled us to obtain detailed data on the butterflies’ flight. Before their operation, the high-speed cameras were calibrated using a portable calibration frame with eight markers. To reconstruct the wing deformation and butterfly flight kinematics, 3 points of the body and 8 individual characteristic points on the fore and hind wings were marked (Figure 1b). The three points on the body were the top of the head, the point between the thorax and the abdomen, and the bottom of the abdomen. The total of sixteen points, located on the wing edge, wing root, wing veins, and characteristic stripes of the fore and hind wings of the butterfly, are shown in Figure 1b. The corresponding three-dimensional coordinate values were obtained by using commercial tracking software (DIPP-MotionV) to establish a three-dimensional global system and the flight kinematics of the butterfly (Figure 1a).

### 2.2. Flow Visualization

To gain a deeper insight into the flow structure of a butterfly, a particle image velocimetry (PIV) system was constructed to record the flow field surrounding the butterfly’s flight. The biological experiment in Section 2.1 and the PIV experiment were independent experiments. The instruments of the PIV experiment included a laser system (Sprout-G, Light-house Photonics Inc. San Jose, CA, USA), a spectroscope, a cylindrical lens, and reflectors. The cylindrical lens formed a laser beam with a wavelength of 532 nm into a light sheet that lightened the specified stationary flow field. Parts of the area might be shaded when the wings of the butterfly pass through the light sheet. This disadvantage might cause inaccuracies in the calculated flow field; thus, in order to solve this problem, a spectroscope was used to divide the laser source into two beams. The two laser beams were projected toward each other on the setting of three reflectors (Figure 2). The beams passed through the separate cylindrical lens, overlapped each other, and formed a single light sheet. The apertures were set between the reflectors and the cylindrical lens to adjust the thickness of the light sheet to be about 1 mm in the experiment.

Experiments on PIV in biomimetic studies are often conducted on an object fixed at a particular point, such as a flapping mechanism [25,35,36] or an actual wing [18,33,37]. For analyses of the flow field of a real hovering insect, a living specimen is often fixed at a point [18,26,33]. The flow structure might differ since the insect may try to break free and thus display a varied wing motion if it becomes stuck. According to previous studies, there has been little research on the flow field of a real hovering butterfly. We endeavored and took on a challenge in the current study. A high-speed camera (Phantom v7.3) operated at a sample rate of 1000 Hz with a resolution of 1024 × 768 pixels was used to record the variation in the flow field. The post-image process was implemented with commercial software (Insight 4G). Alumina powders with a particle diameter of 3 μm were chosen as the tracer particles. A tracer particle was considered appropriate when the settling velocity, denoted as $u_{\infty }=$
*gd*_p_^2^(*ρ*_p_ − *ρ*_f_)/18*μ*, was much lower than the flow velocity [38], where *ρ*_p_ and *ρ*_f_ are the particle and fluid densities, respectively; *d*_p_ denotes the diameter of a particle; *g* denotes gravity; and *μ* denotes the fluid viscosity. The value of the settling velocity was of order 10^−3^–10^−4^, which was much smaller than the flow field velocity; the tracer particle was hence suitable for the experiment.

### 2.3. Simulation Scheme and Wing Model

Numerical analyses were applied with commercial software (ANSYS FLUENT) to address the flying behavior of a butterfly. The finite-volume method was used, and the semi-implicit method for pressure-linked equations consistent (SIMPLEC) was adopted to solve the pressure and velocity fields. The numerical calculation was solved with two workstations (AMD Ryzen Threadripper PRO 3995WX, 64-core 2.70 GHz, RAM-128 GB; AMD Ryzen Threadripper 3990X, 64-core 2.90 GHz, RAM-128 GB). The governing equations are
(1)$$\nabla \cdot \vec{u}=0$$
(2)$$\rho_{f}\left(\frac{D\vec{u}}{\mathrm{Dt}}\right)=-\nabla P+\mu \nabla^{2}{\;\vec{u}+\rho }_{f}\vec{g}$$
in which $\vec{u}$ denotes the velocity vector, *t* denotes time, $\vec{g}$ denotes gravity, and *P* is the pressure field. The density of the fluid was set to *ρ_f_* = 1.23 (kg/m^3^), and the fluid viscosity was set to *μ* = 1.79 × 10^−5^ (Pa s). The shape of the wing model (Figure 3a) followed the real size of a butterfly (*Tirumala septentrionis*). The morphological parameters were obtained from 4 individual butterflies (*n* = 4), and the parameters of the simulation model are shown in Table 1. Thirteen groups of butterfly flight videos were recorded, with an average of nearly three videos per butterfly. Afterward, the videos were analyzed to examine the relationship between flight speed and body angle. Each group of data represented a single flapping cycle. As the fore and hind wings of a real butterfly partially overlap [8], we assumed the wings to be broad wings. The thickness of the wing model was 2.5% of the mean wing chord, which is a better aspect ratio of the mesh based on our previous study [22]. According to previous studies [39,40], the wing rotation radius (*r*_2_) can better reflect the relationship between the aerodynamic force of the wings, the wing shapes and sizes, and the vortex at the leading edge of the wing. The radius of gyration was 24.51 mm, which was 53% of the length of the wingspan. The power was calculated as *P* = $\sum_{i=0}^{n}{\vec{F}}_{i}\;\cdot {\;\vec{v}}_{i}$, where *n* is the total number of grids on the wings and ${\vec{F}}_{i}$ and ${\vec{v}}_{i}$ represent the aerodynamic forces and velocity on each element, respectively. The coefficients of thrust (*C_T_*) and lift (*C_L_*) were normalized according to the mean wing-tip velocity, which is consistent with the methodology used in previous studies [12,19,22].

As the Reynolds number of a flying butterfly is about 6 × 10^3^, an incompressible and laminar fluid flow was assumed. The butterfly model was placed in a spherical domain of 15S in incompressible air at 25 °C, in which S equals the wingspan of 48 mm. The fluid domain was divided into four regions. The inner spherical diameters were 2S, 4S, 10S, and 15S (Figure 3b). The outside boundary was the pressure outlet, *P* = −*ρgy*; the surface of the wing model was set to a no-slip condition. The negative Y-axis direction was gravity; the XZ plane was the ground. Each flapping cycle comprised 400 time steps. To obtain accurate flow field information, the density of the mesh grids was enhanced in the inner region (1.2 mm mesh grids) and the second-layer region (Figure 3b); the face sizing around the wing model was set (0.4 mm mesh grids) to obtain precise solutions (Figure 3a). Figure 3c shows that the lift converges into three distinct grid numbers. The three different grid sizes were adjusted by reducing the grid size to increase the resolution of the computational domain. The relative difference in the average lift force between the cases of fine and medium mesh grids was less than 1%. The total mesh number of 9.0 × 10^6^ was used with tetrahedral grids in this work; a dynamic mesh with a method of smoothing and remeshing was also applied.

To better design MAVs, it is considered that the flapping motion of a butterfly is one-dimensional. However, many studies have suggested that the wing motion of a butterfly is a simple trigonometric function [41], which does not parallel a butterfly’s real flight with asymmetric flapping [18,22,42]. To approach the butterfly’s real flight, the equations published in the literature [22] were used to calculate the butterfly’s motion, including the body angle, flapping motion, wing-pitch motion, and deviation angle. The flapping motion and the body angle were measured between the center line of the thorax and the horizontal plane of the butterfly. The vertical and horizontal body angles represented 90° and 0°, respectively. The wing-pitch motion was depicted in the simulation, as shown in Figure 3d. Nevertheless, the deviation angle was disregarded because its impact was relatively lower than that of the body angle and wing flexibility, as indicated by a previous study [12]. The flapping motion and the body angle at the low horizontal velocity of 0.42 m s^−1^ were calculated, as shown in Figure 3e and Figure 3f, respectively. The fitting curve of the wing motions was calculated by using the discrete Fourier transform (DFT; *n* = 3), and it is displayed as a red dashed line. These data were then used to prescribe the wing motions and body angles in the simulation.

### 2.4. Kinematics and Deformation of the Wing Model

To understand the magnitude, change in lift between the wing flexibility and the difference in body angle, three fixed body angles—the minimum, average, and maximum— were chosen to calculate the lift between the body angle and the camber deformation of the butterfly. As butterfly wings consist of a membrane and veins, forming an anisotropic structure, the measurements of Young’s modulus and Poisson’s ratio are challenging. Therefore, this study adopted a predefined deformation method to carry out numerical simulations. The camber deformation was defined as the vertical displacement (*y_w,camber_*) based on the wing plane *x_w_y_w_z_w_*, varying in chordwise planes, as shown in Figure 4. The chordwise plane is the vertical plane of the connection between the leading edge and trailing edge (Figure 4a). Figure 4a shows the minimum and maximum cambers in the chordwise plane. The camber deformation is dependent on the chordwise direction, the value of which varies with the spanwise position and time. We collected one set of data specifically for low speed flying with the aim of reconstructing the camber deformation (Figure 4b). The data corresponded to a horizontal velocity of 0.42 m s^−1^. The blue and green solid lines in Figure 4b indicate the averages of the maximum and minimum cambers in all chordwise planes, respectively, and the blue and green regions represent the standard error of the mean (SEM). The wing deformation was distinctly convex and concave during downstroke and upstroke flapping, respectively. The solid black line in Figure 4b represents the maximum and minimum values of each convex and concave deformation. Afterward, the discrete Fourier transform (DFT; *n* = 8) was employed to calculate the data and to fit the characteristic camber, which is shown as a red dashed line. The fitting curve was used to prescribe the characteristic camber in the simulation, while an algorithm with a user-defined function (UDF) was established to control the position of each mesh grid of the wing model for simulating the wing deformation. The value of the deflection of the wing was calculated based on the difference between the deformation and the reference planes. The UDF was applied to systematically control the grids instead of specifying the amount of displacement on each mesh grid. The wing’s characteristic camber deformation (wing chordwise displacement) was determined using a quadratic function (3):*y_w_*(t) = *a*(t)·*x_w_*^2^ + *b*(t)·*x_w_* + *c*(t)(3)

Equation (3) is a three-dimensional surface: *w* is the wing coordination; *x* is the chord position; *y* is the grid displacement (unit: mm) on the wing chord; and *c* is an arbitrary number. The reference plane was formed by connecting the leading edge (LE) and the trailing edge (TE), constituting the longest chordwise length. The fixed parameters *l*_1_ and *l*_2_ represent the distances between LE and TE to the wing root line (*z_w_*), respectively, as shown in Figure 4c. To ensure that there is no deformation at the wing root (*x_w_* = 0), LE and TE were moved down a characteristic camber (*h*(t)) as deformation, and the wing root coordination became (0, 0). The arbitrary number *c*(t) was also eliminated in Equation (4) after the substitution of the LE (*l*_1_, −*h*(t)) and TE (−*l*_2_, −*h*(t)) coordinates and the characteristic camber into Equation (3). The two coefficients *a* and *b* also varied with time, and the characteristic camber in Equation (4) mainly changed with the camber deformation. In the simulation, the wing model deformed as a curve along the spanwise direction. The shape and scale of the deformation in the spanwise direction were similar, resulting in the formation of a 3D wing. The wing deformation of the three-dimensional surface was increased or reduced by multiplying coefficient *K* (Figure 5) and characteristic camber (*h*(t)). Afterward, the coordination of the three points was brought into the equation; the arbitrary number *c* was eliminated, and it became the following:(4)$$\left\{\begin{array}{c} a(t)\cdot l_{1}^{2\;}{+\;b(t)\cdot l}_{1}=K\cdot -h(t)\\ a(t)\cdot l_{2}^{2\;}-{\;b(t)\cdot l}_{2}=K\cdot -h(t) \end{array}\right.$$

According to the function curve calculation, the wing becomes a deformable three-dimensional surface, varying with time periodically.

## 3. Results and Discussion

### 3.1. Biological Observation

The observation data of the body angle with forward flight speed during the butterfly flight are plotted in Figure 6a. Each black circle in the figure represents the values of the body angle and forward flight speed recorded in image 0.001 s^−1^ with the high-speed cameras. Each group of videos recorded during the butterfly flight generally contains 80 to 110 measurements, and the red circle in each group’s data represents the average value, ranging within 10° to 60° of the body angle of the 13 groups of data. The maximum body angle during the butterfly flight was 75.41°, of which the horizontal velocity was 0.42 m s^−1^, and this speed was the lowest among all horizontal velocities. The trend shown in Figure 6b indicated that the decreased average body angle would increase the forward flight speed from 0.42 to 1.33 m s^−1^. The coefficient of determination of the linear regression was 0.847, which revealed a high level of relevance. Overall, the results found in the present study correspond well with those of many studies, indicating that butterflies alter their body angle to change the wing flapping direction and increase the horizontal aerodynamic forces [11]. In addition, other studies have asserted that it is not only the body angle that can generate more lift but also the flexible wing [25,26,27]. It is thus worth investigating how much lift can be generated when the body angle and wing deformation change during butterfly flight.

### 3.2. Experimental Results of Particle-Image Velocimetry on a Low-Speed Butterfly

To analyze the flow field conditions of a butterfly flying at low speeds, PIV was used, and the flow velocity of the fluid particles was calculated. Figure 7 shows a flow visualization of a butterfly with a low horizontal velocity recorded with a high-speed camera, and the vorticity contour was calculated with Tecplot software 2021 R1. At the beginning of the downstroke (*t** = 0.11), the vortices detached around the left and right wings of the butterfly (Figure 7a). These vortices constituted a three-dimensional vortex ring that had formed in the upstroke of the preceding cycle and displayed two opposite vortices in a two-dimensional plane. The vortex ring continuously rotated toward the center and produced a central jet flow inside the ring [19], as indicated by an arrow. At the beginning of the upstroke (*t** = 0.54), besides the detached vortices left in Figure 7a, the vortices generated by the left and right wings detached and remained around the wings. The two vortices also rotated clockwise and counterclockwise toward each other, inducing a fluid jet to flow downward between those vortices (Figure 7b). Figure 7c shows the flow field of the mid-downstroke in the next cycle (*t** = 1.25) around the butterfly with many detached vortices. The vortices at the bottom formed and lasted for some time, as shown in Figure 7a, but they were going to dissipate. The detached vortices of the previous cycle also formed beneath the wing. In the flow visualization experiment, the flow field conditions of a butterfly with a low horizontal velocity were observed. The body of the butterfly was nearly perpendicular to the ground, and it flapped its wings forward and backward. In theory, this should produce jet flow vortices in either the forward or the backward direction. However, the flow visualization revealed the detachment of the vortex flow toward the downward direction (Figure 7). Each stroke generated a pair of vortices from the left and right wings, and each pair of vortices rotated in opposite directions, continuously producing a downward jet flow to support the body weight at low horizontal speed. The jet flow was strongly induced by the wings during the downstroke and upstroke. This phenomenon may be largely affected by the body angle and chordwise deformation [11,27]. In the next section, numerical simulations were conducted to decouple these two effects.

### 3.3. Simulation Results

The flying trajectory of butterfly flight is erratic. It is thus challenging to deduce the general features of a cycle of wing motion from the experiment. Thus, a numerical simulation was adopted to study the lift generated from the chordwise deformation and the body angle during butterfly flight. A characteristic camber of the wing was chosen first, and then cases with amplified deformation were studied. The amplitudes of *K* were set to 0 and 3, corresponding to the rigid wing and the varied camber wing, respectively, whereas the body angle was fixed at the average value (55.01°) of a flapping cycle for the simulation processes.

The results of the force distribution and the overall wing performance are shown in Figure 8 and Table 2. The total force was calculated by averaging the entire cycle. Figure 8a shows that the lift and the trends of variation were quite similar during the downstroke, while the negative lift decreased with an increasing body angle during the upstroke. The trends were quite similar during the flapping cycle, but the thrust values had some differences among *K* = 1–3 (Figure 8b). Table 2 shows the overall force, power, lift-to-power ratio (*L*/*P*), thrust-to-power ratio (*T*/*P*), and total force-to-power ratio of a flapping cycle and the percentage variations in the different amplitudes of *K*, which were normalized to the real butterfly characteristic camber *K* = 1. The mean lift increased with an increasing body angle, and a similar result was found for the mean thrust when *K* < 3; yet the mean thrust declined when *K* = 3. In contrast, the total force and power decreased with increasing *K* due to the more positive values on the lift and the thrust. This result illustrates that the characteristic camber wing has more LEV attachment during flight, enhancing the wing’s performance more efficiently.

A previous study [11] indicated that the thrust generation during the upstroke is significant for propelling a butterfly forward in flight. However, the study used the rigid wing as a numerical simulation model, which might not accurately reflect the flexibility and aerodynamics of butterfly wings in real conditions. In this work, the mean thrust variation between *K* = 0 (rigid wing) and *K* = 1 differed significantly, by 45.6%, which was attributed to the relatively large drag of the rigid wing, causing more negative thrust during the downstroke. The comparison reveals that using the rigid wing model in simulations may have limitations in accurately predicting real-world conditions.

Table 2 and Figure 8b show that differences in the average values of thrust exist between *K* = 2 and 3 during the upstroke. When *K* = 3, the mean thrust decreased during the upstroke, but the lift still increased. When *K* = 2, the cambered wing generated less negative thrust and negative lift during both the downstroke and upstroke. The thrust, however, decreased when *K* = 3, and the pressure distribution on the wing was almost the same as when *K* = 2 during the upstroke. It is worth noting that *K* = 3 had more camber deformation during the upstroke, as shown in Figure 4b and Figure 9. The curvature of the leading edge of the wing diverted the horizontal force to the vertical direction during the upstroke, resulting in the jet flow being directed downward. This effect was also found in a previous study [43].

An increase in *K* enhances the mean lift, as mentioned earlier. In short, the relation between the direction and magnitude of the jet at the end of the flapping cycle is shown in Figure 10. LEV_u_ (where u indicates the vortex formed in the upstroke) and TEV_u_ merged on the ventral side of the wing. LEV_u_ detached and combined with WTV_u_ (wing-tip vortices) and formed a complex structure of LEV_u_-WTV_u_ at the wing tip. These detached vortex structures formed a vortex ring and played an important role in generating lift and thrust during flight, similarly to those in previous studies [11,12,19,20,44]. This implies that the stable attachment of LEV leads to a strong and efficient vortex structure, resulting in the wing producing a more conducive force [12,19,26]. Figure 10a–d show that increasing the *K* value results in more jet flow and a greater downward force during upstroke and downstroke, leading to increased velocity generation. The figures also depict detached vortices during upstroke and downstroke, and it appears that the vortex size increased slightly as the *K* value increased. The phenomenon of an increasing downward jet flow is similar to that of an increasing body angle, and it also resembles the PIV experience mentioned in Section 3.2.

The same wing deformation, i.e., *K* = 1, the real butterfly camber deformation, was used to examine the lift effect at different body angles. The data from measuring the body angle at the maximum, average, and minimum during the flapping cycle, i.e., 75.41°, 55.01°, and 39.41°, respectively, were input into the simulation. The simulation results indicated that the lift magnitude substantially increased during the downstroke and that the negative lift magnitude also significantly increased during the upstroke when the body was at a lower angle (Figure 11a). These simulation trends agree well with previous studies [26,27]. Table 3 shows the parameter variation simulated with different *K* values. The table also shows the percentage variations in the maximum and minimum values normalized to the average value of the body angle. The percentage difference between the thrust and *T*/*P* at 39.41° varied significantly and could reach approximately 45 %. The corresponding values at 75.41° differed most substantially and reached around −73%. The variations in the mean lift and lift-to-power ratio at both angles were relatively minor and reached approximately −18% and 12%, respectively. The percentage differences in the other parameters were relatively minor, generally <2%. The mean lift increased with an increase in the body angle, and vice versa for the mean thrust, resulting in a change in the flapping direction. Figure 12 further shows that the vortex ring direction at the body angle of 39.41° was backward, and at 75.41°, it was bottom left at the end of the upstroke. This difference in body angle caused the butterfly to produce a more effective lift when the body angle was more perpendicular to the ground.

The thrust variation was significant during the downstroke between two different body angles, 39.41° and 75.41°, while the values varied in a relatively minor manner during the upstroke, as shown in Figure 11b. Figure 13a–c reveals that, when the body angle increased, the force projected in the vertical direction gradually reduced, and vice versa for the horizontal direction during the downstroke, resembling the results in Figure 11. Figure 13d–f illustrates that the force projection in the vertical direction markedly varied from downward to upward and turned the lift from negative to positive. In contrast, the projection of the horizontal force remained fairly constant and was not relevant to the body angles, as shown in Figure 11b at the normalized time of 0.7.

The trend of the average lift coefficient at the three *K* coefficients and body angles in Figure 14 is consistent with the results in Table 2 and Table 3. The mean lift increased with coefficient *K* and the ascending body angle. The highest average lift coefficient was observed at *K* = 3 for the various body angles, while the average thrust coefficient nearly exhibited the opposite trend to the average lift coefficient, as indicated in Figure 14b. All camber deformations at *K* = 2 had the greatest average thrust coefficient at the different body angles. However, all values at *K* = 3 showed a decreasing trend. The power decreased as the *K* coefficient increased, and the results indicate that the cambered wing performed more efficiently during flight. The lift coefficient at *K* = 3 at a lower body angle (39.41°) was nearly equal to *K* = 1 at an average body angle of 55.01°. A similar result was also seen for *K* = 3, where the average body angle (55.01°) nearly paralleled *K* = 1 at a body angle of 75.41°. All these results suggest that a wing with a significant camber deformation could generate the same lift coefficient as a wing with a higher body angle. The camber deformation at *K* = 3 could increase lift by shifting the horizontal thrust to the vertical direction, as mentioned above. A similar result can be seen in Figure 12, which illustrates that the higher the body angle, the lower the average thrust coefficient.

## 4. Conclusions

In the present work, we carried out a series of biological experiments and a numerical simulation to reveal the lift variation induced by body angle changes and chordwise deformation during the flight of a butterfly (*Tirumala septentrionis*). The biological observation experiment aimed to investigate the effects of body angle and wing deformation on flight speed. The results show that decreasing the body angle increases the forward flight speed, which is well consistent with many previous studies. An analysis with particle image velocimetry was also conducted to quantitatively reveal the flow field and the vorticity of the transient flow field around the butterfly.

To decouple the effects of the body angle and wing deformation on the lift of the butterfly, a systematic numerical simulation was carried out to examine the force generation of the wing when fixing three body angles at the minimum (39.41°), average (55.01°), and maximum angle (75.41°), respectively. The value of camber deformation was calculated with the maximum value of chordwise deformation, which was brought into a quadratic function to represent the characteristic camber. The chordwise deformation was amplified by increasing or decreasing the multiplication of coefficient *K*. The obtained results indicate that it is not only the body angle but also the wing flexibility that can enhance lift. The latter generates a more efficient lift during the upstroke. The simulation results show that both the mean lift and the mean thrust generally increase as *K* increases. The total force and power decreased with increasing *K*, and when *K* = 2, they decreased by at least 5% and 8%, respectively. However, when *K* = 3, the mean thrust decreased, which might be attributed to the vortex attaching to the curvature of the leading edge of the forewing and causing the horizontal force to shift toward the vertical direction during the upstroke. Moreover, a significant variation in wing camber deformation (*K* = 3) was found to generate the same lift coefficient as a higher body angle. The lift coefficients increased by 18.4%, 14.1%, and 10.4% from the minimum, average, and maximum angles when *K* = 3 compared to *K* = 1. This study indicates that, beyond the body angle, chordwise deformation also plays a significant role in generating lift and altering the force direction during butterfly flight. However, an important consideration in this study is the use of a one-way fluid-structure interaction simulation, as mentioned earlier, due to the anisotropic structure of butterfly wings. Real butterfly wings also exhibit deformations with camber and twist variations. In our study, we focused on the impact of camber deformation and body angle on lift through a parametric analysis.

With the inspiration of real butterfly flight, the lift magnitude of a butterfly was found to be generated by both the body angle and chordwise deformation. This finding could provide insight for microaerial vehicle design. Wing deformation combined with body motions might also significantly influence the aerodynamics of butterfly flight, which we will examine in future research.

## Figures and Tables

**Figure 1 biomimetics-08-00287-f001:**
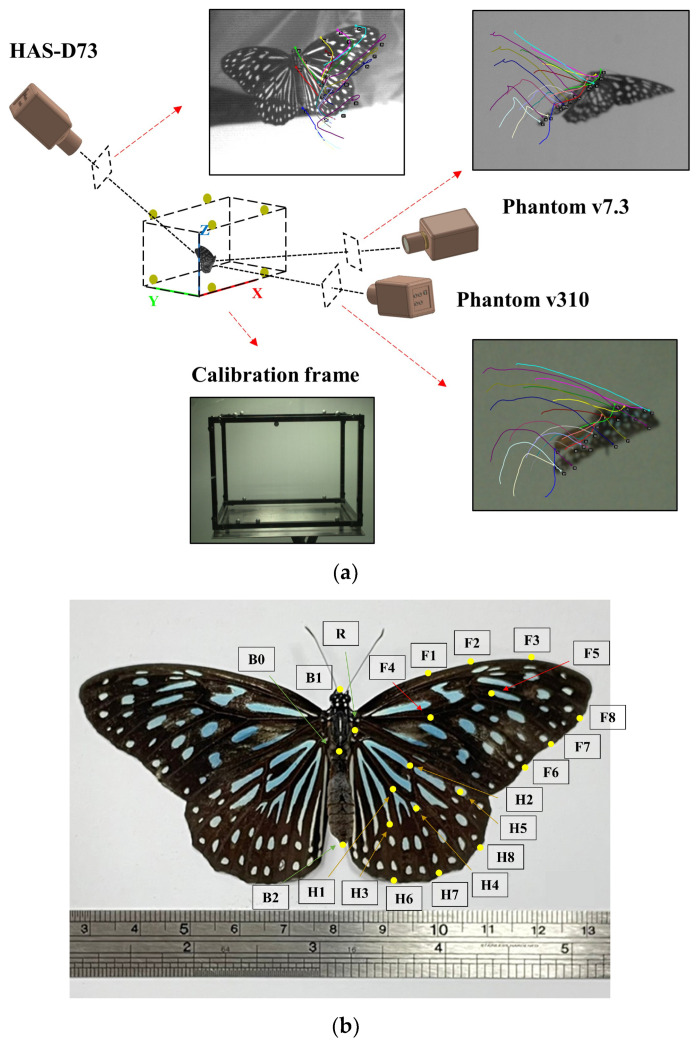
(**a**) Three high-speed cameras record the real free-flying butterfly, and a calibration frame is used to construct the three-dimensional coordinates. (**b**) The characteristic points on the butterfly’s body and wings.

**Figure 2 biomimetics-08-00287-f002:**
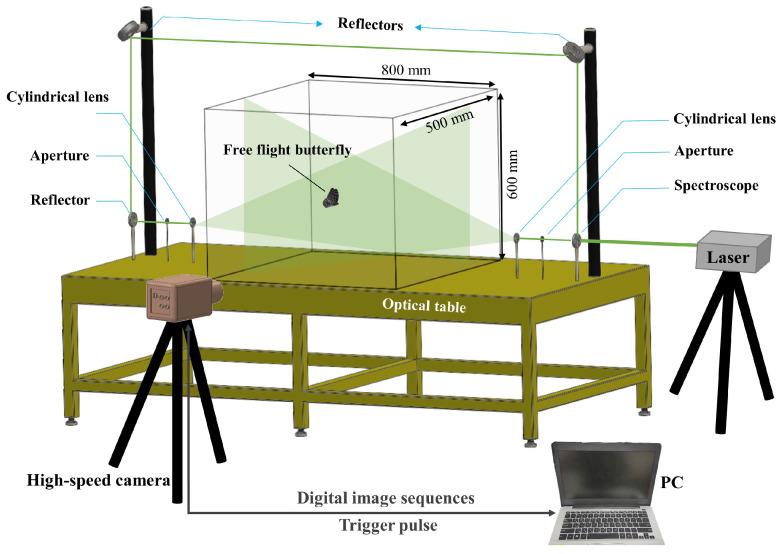
Experimental setup of the particle-image velocimetry (PIV).

**Figure 3 biomimetics-08-00287-f003:**
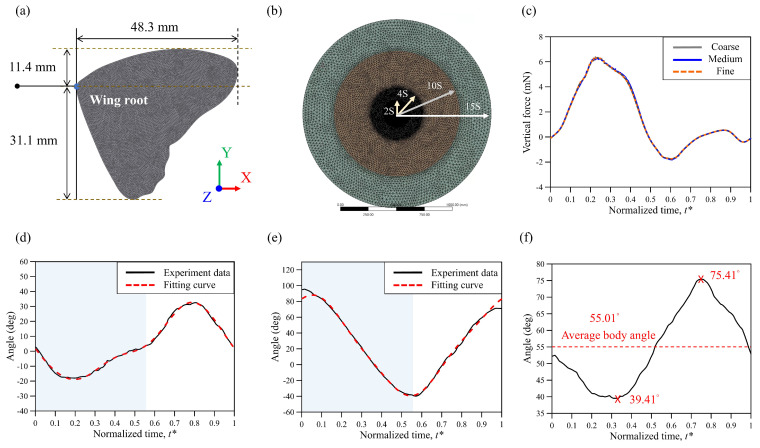
(**a**) Size of the wing model with face sizing; (**b**) the fluid domain was divided into four regions of varied grid numbers. The inner region had the greatest mesh grid density; (**c**) grid convergence test among coarse grid (4.2 million), medium grid (9.1 million), and fine grid (14.1 million). (**d**) The wing-pitch motion; (**e**) the asymmetric flapping motion, where the transfer of the downstroke and upstroke occurred at *t** = 0.55. The downstroke period is depicted on a light blue background. (**f**) The variation in the body angle throughout a cycle and three specific body angles are indicated.

**Figure 4 biomimetics-08-00287-f004:**
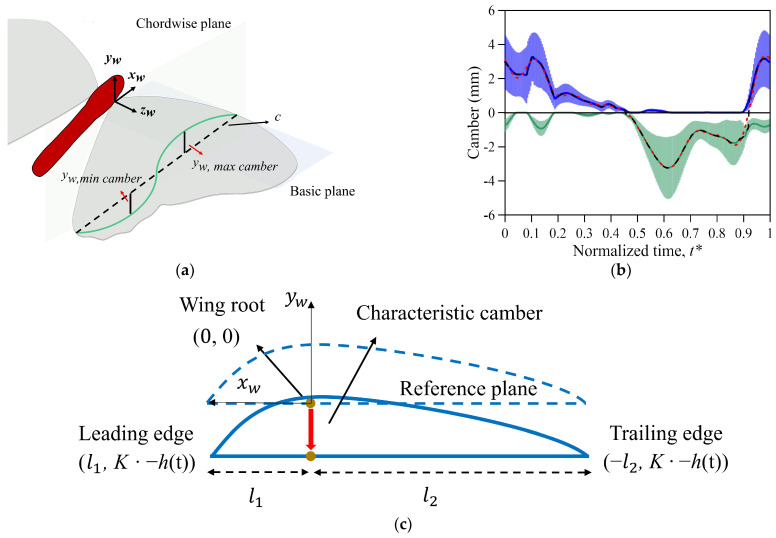
(**a**) The definition of the camber on the chordwise plane. (**b**) The maximum camber and minimum camber deformation values. The blue and green solid lines in the figure indicate the averages of the maximum and minimum cambers in all chordwise planes, respectively, and the blue and green regions represent the standard error of the mean (SEM). (**c**) A schematic diagram of characteristic camber deformation in the view of the spanwise direction.

**Figure 5 biomimetics-08-00287-f005:**
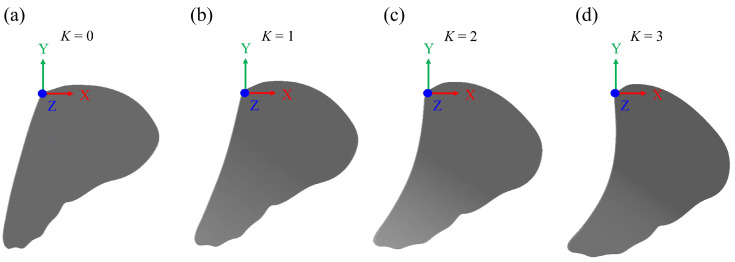
The different amplitudes of the wing deformation are denoted by the coefficient *K*. (**a**) *K* = 0 (rigid wing). (**b**) *K* = 1. (**c**) *K* = 2. (**d**) *K* = 3. (*t** = 0.6).

**Figure 6 biomimetics-08-00287-f006:**
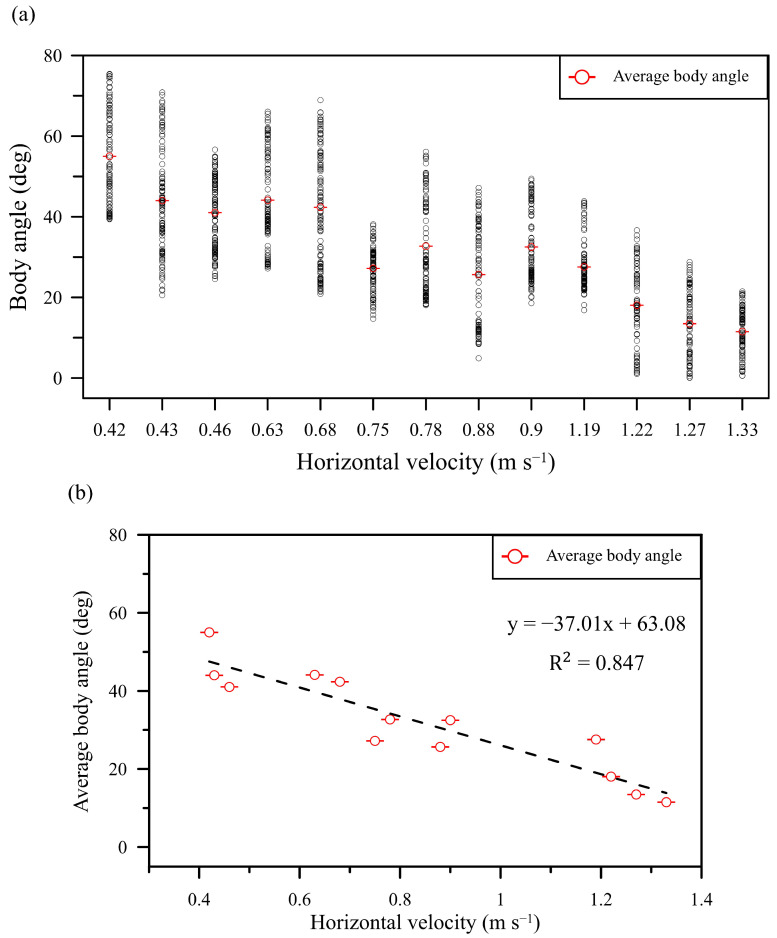
(**a**) The relationship between body angle and forward flight speed with the horizontal velocity from slow to fast, and (**b**) the linear regression of the average body angle and horizontal velocity.

**Figure 7 biomimetics-08-00287-f007:**
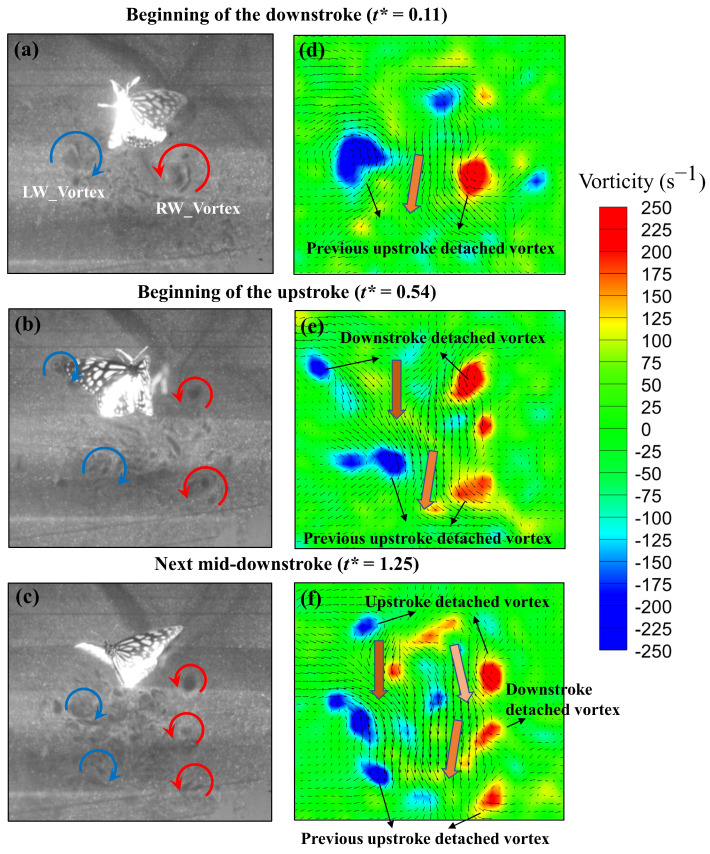
Flow visualization of a hovering butterfly and the calculated flow field and vorticity at (**a**) the beginning of the downstroke, (**b**) the beginning of the upstroke, and (**c**) the next mid-downstroke. (**d**–**f**) represent the calculated vorticity in contrast to (**a**–**c**), respectively.

**Figure 8 biomimetics-08-00287-f008:**
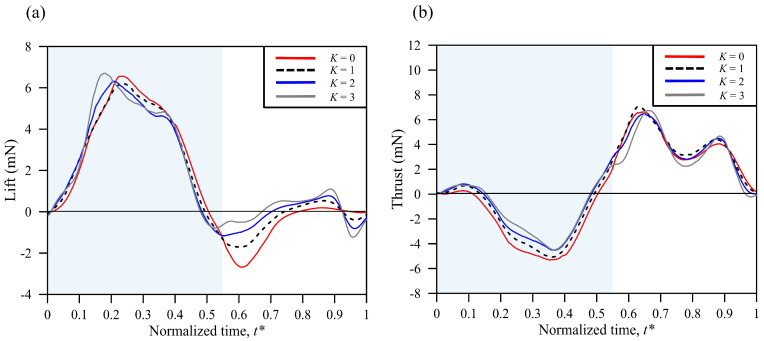
The different amplitudes of the wing deformations of (**a**) lift and (**b**) thrust in the simulation. The light blue background represents the downstroke period, while the rest represents the upstroke period.

**Figure 9 biomimetics-08-00287-f009:**
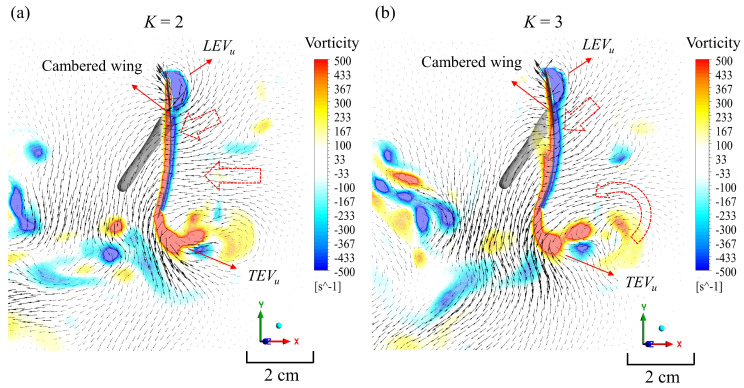
Vorticity and velocity vectors of (**a**) *K* = 2 and (**b**) *K* = 3 are shown in the plane of position r^2^ at the middle of the upstroke (*t** = 0.75). The red arrows indicate the change in the vector direction as the wing undergoes a more cambered formation.

**Figure 10 biomimetics-08-00287-f010:**
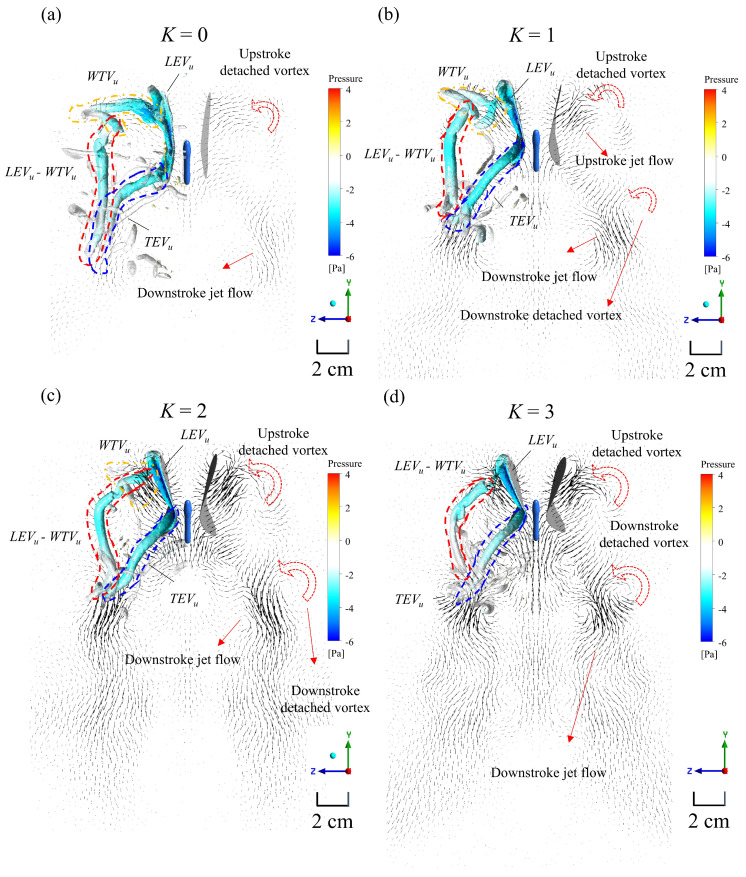
The velocity vectors of (**a**) *K* = 0, (**b**) *K* = 1, (**c**) *K* = 2, and (**d**) *K* = 3 are shown in the plane at 50% wingspan at the end of the flapping cycle (*t** = 1). The red arrows indicate the change in both the vector direction and position as the wing undergoes a more cambered formation.

**Figure 11 biomimetics-08-00287-f011:**
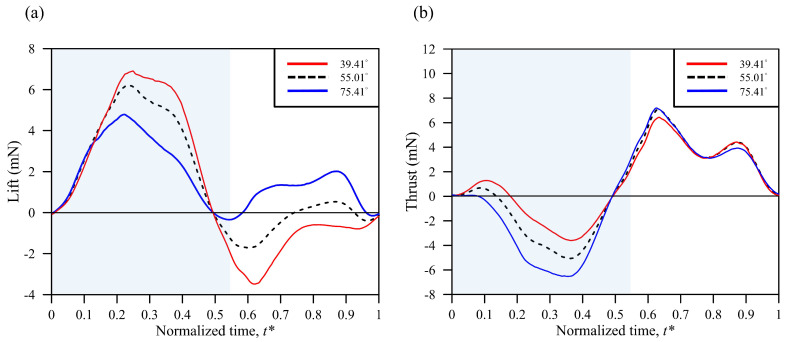
The characteristic camber *K* = 1 with different body angles of (**a**) lift and (**b**) thrust. The light blue background represents the downstroke period, while the rest represents the upstroke period.

**Figure 12 biomimetics-08-00287-f012:**
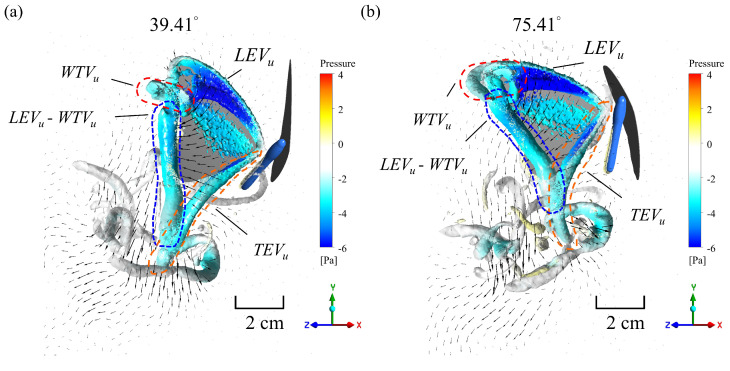
Velocity vector and pressure contour of three-dimensional vortex structure in cases with (**a**) a body angle of 39.41°and (**b**) a body angle of 75.41° at *t** = 0.9. The *Q* criterion is used to determine the vortex structure: *Q* = 30(2Δ*ϕf*)^2^ = 80,870 s^−2^, which is 30 times the mean flapping angular velocity.

**Figure 13 biomimetics-08-00287-f013:**
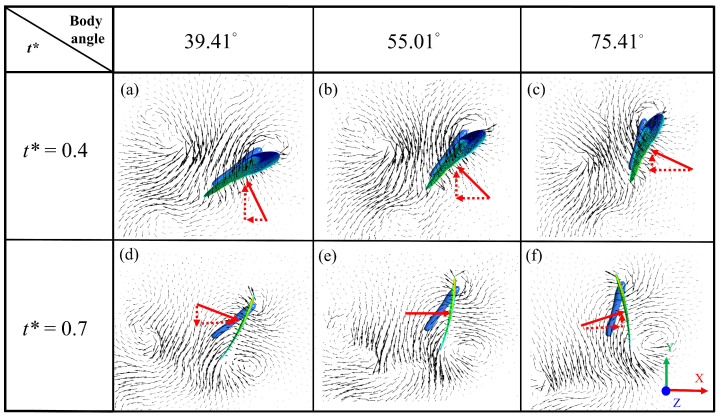
A schematic diagram of the force direction on the wing at the different body angles and the two different normalized times, 0.4 and 0.7. (**a**–**f**) represent the corresponding body angles over time. The colored area on the wings represents the pressure distribution. The red arrow indicates the force on the wing’s normal vector, and the red dash arrow represents the force projection to the horizontal and vertical directions during downstroke and upstroke.

**Figure 14 biomimetics-08-00287-f014:**
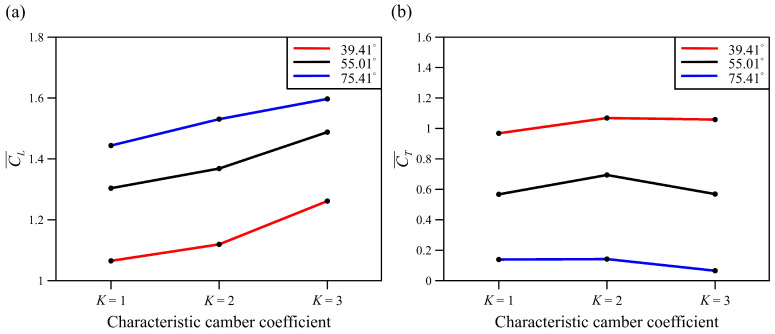
The average lift (**a**) and thrust (**b**) coefficient on the wing at different *K* coefficients and body angles.

**Table 1 biomimetics-08-00287-t001:** Biological morphology of the wing model in simulation based on experimental data.

	Simulation	Experiment (Average ± SEM)
Total mass (g)	0.35	0.36 ± 0.04
Flapping frequency (Hz)	11.00	10.77 ± 0.38
Wingspan (mm)	48.00	47.50 ± 0.63
Mean wing chord (mm)	20.11	19.79 ± 0.14
Aspect ratio	2.4	2.40 ± 0.03

**Table 2 biomimetics-08-00287-t002:** The wing aerodynamics are based on the different *K* values, and the body angle is set to the average body angle of 55.01°.

Wing Flexibility	Mean Lift (mN)	Mean Thrust (mN)	Total Force (mN)	Power (mW)	L/P (N/W)	T/P (N/W)	F/P (N/W)
*K* = 0	1.48	0.449	5.73	7.44	0.198	0.060	0.770
	(−6.9%)	(−45.6%)	(4.2%)	(6.6%)	(−13.2%)	(−49.2%)	(−2.3%)
*K* = 1	1.59	0.825	5.50	6.98	0.228	0.118	0.788
*K* = 2	1.67	0.892	5.22	6.30	0.265	0.141	0.829
	(5.0%)	(8.1%)	(−5.1%)	(−9.7%)	(16.2%)	(19.5%)	(5.2%)
*K* = 3	1.81	0.839	4.92	5.90	0.307	0.142	0.834
	(13.8%)	(1.7%)	(−10.5%)	(−15.5%)	(34.6%)	(20.3%)	(5.8%)

**Table 3 biomimetics-08-00287-t003:** The wing aerodynamics based on the different body angles; *K* was set to 1 to reflect the real butterfly camber.

Body Angle	Mean Lift (mN)	Mean Thrust (mN)	Total Force (mN)	Power (mW)	L/P (N/W)	T/P (N/W)	F/P (N/W)
39.41°	1.30	1.19	5.56	6.92	0.188	0.172	0.305
	(−18.2%)	(44.2%)	(1.1%)	(−0.9%)	(−17.5%)	(45.8%)	(0.3%)
55.01°	1.59	0.825	5.50	6.98	0.228	0.118	0.304
75.41°	1.77	0.224	5.59	6.94	0.255	0.032	0.307
	(11.3%)	(−72.9%)	(1.6%)	(−0.6%)	(11.8%)	(−72.9%)	(1.0%)

## Data Availability

The data that support the findings of this study are available from the corresponding author upon reasonable request.

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
