# Peer review of "The Lift Effects of Chordwise Wing Deformation and Body Angle on Low-Speed Flying Butterflies"

_biomimetics, 2023, doi:10.3390/biomimetics8030287_

Round 1

Reviewer 1 Report

In their manuscript, the authors quite clearly formulated and proved the main idea about the effect of wing deformations and body angle on the generation of aerodynamic forces.

There are a few issues, which, in my opinion, should be corrected:

1. There are no data about sample. It is necessary to indicate how many individuals were recorded and measured, how many wing cycles were processed and analyzed. In the Table 1 authors presents parameters of insects. Are the distributions normal? Can SEM be used to describe these samples?

2. Quality of figures is poor because of the low resolution. It would be better to use vector graphics for most of figures and 300ppi or higher for raster elements.

3. Lines 193, 196, 216. Figure references are incorrect (3a-c instead of 4a-c).

4. Figure 6. I think that it would be much clearer to display not individual data points, but average values for kinematic cycles. Because the angle of the body changes during the cycle.

5. Line 240-242. There are no trend on Figure 6. It is necessary to carry out a statistical analysis to show the dependence of the flight speed on the body angle.

6. Figures 8 and 11. I propose to labeling the phases of the wingbeat cycle in the figures (downstroke and upstroke).

7. Table 2. How was the total force calculated? Is this the average force over the cycle or the maximum? How was the power calculated? Is it average aerodynamic power or aerodynamic plus inertial power? I propose to clearly describe the calculation of these characteristics in methods or results section.

8. Authors too often use the wording: "Figure or Table N shows... ". It would be better to paraphrase these passages in the text to improve its readability.

I recommend publication after improving methods, results and figures.

Author Response

Please find the attached file (20230616 biomimetics-2429893 Respone to comments of reviewers_Fang).

Reviewer 2 Report

Flight of butterflies is an outcome of very complex coupled dynamics. As such modeling and analysis is very challenging. This manuscript provides experimental measurements of the motion as well as some PIV measurements and CFD results of the flow around butterflies. In this regard I regard this study as a welcome addition to the literature.

Nevertheless, I have some concerns that I think need to be addressed before I can recommend acceptance of this manuscript in this journal:

1. In general an abstract should provide a succinct highlight of the employed study. However the abstract in its current form is difficult to follow. For example:

  • “Systematic numerical simulations were …” It is unclear what is meant by “systematic”. Please be more specific.

  • “The results show that the chordwise deformation enhances wing performance by increasing the strength and stabilizing the vortex attachment.” Does the chordwise deformation always lead to the observed vortical behavior?

  • “The results show that the chordwise deformation enhances wing performance by increasing the strength and stabilizing the vortex attachment.” What is meant by the vortex attachment strength?

  • “The wing undergoes a significant chordwise deformation that could generate a lift coefficient equivalent to that with a higher body angle” This sentence is difficult to understand without the insight into the relation between the lift coefficient and the body angle, which is not provided in the abstract.

  • “This increase is due to the leading-edge vortex attached to the curved wing alters the horizontal force to the vertical direction, and thus, produces a more efficient lift during the flapping upstroke.” This sentence needs to be reworded.

2. The description of the camber deformation model is cryptic and difficult to follow. For example, a(t), b(t), c(t), and h(t), which are key values, don’t seem to be reported. 

Moreover, in Lines 221-213, it is stated that “the equation (3) is a three-dimensional surface: w, the wing coordination; i, the mesh number; x, the chord position; y, the grid displacement (unit: mm) on the wing chord; c, an arbitrary number.” How does Eq. (3) represent a 3D surface if there are only two coordinates (x,y)? Also, a comma between w and i will make it more clear. In addition, it is difficult to see how i is defined. A schematic defining i would be helpful, for example in Fig. 4a.

Furthermore, does Eq. (3) hold for the wing root? Substitution of (0,0) results in c(t)=0.

3. Are the wing motion measurements and PIV measurements conducted at the same time? If so, it will be useful to emphasize this as this is a challenging task. If not, it will be useful to note this at the beginning of the Methods section. This way it will be easier for the readers to follow how the experiments are set up. 

4. One major benefit of a combined experimental and numerical study is that the results can be compared against each other. For example, is the UDF of the CFD wing camber deformation validated against the results from the experiments? Also, shouldn’t the CFD and PIV flow field results be compared to each other? A description of which experimental results have been used in the CFD simulations (for example, the flapping angle, etc.) and how will be helpful for the readers to follow.

5.  Limitations of the employed method should be clearly stated and discussed. For example, the used FSI is a 1-way coupling model. The wing deformations affect the force generation, which has been captured. However, the effects of the pressure distribution on the wing deformation have not been modeled. How do these limitations affect the conclusions of this study?

6. Effects of camber deformation on the lift production and vortex dynamics were modeled and discussed in the following reference. This reference should be added to the manuscript.

Kang Chang-kwon and Shyy Wei, 2013, Scaling law and enhancement of lift generation of an insect-size hovering flexible wing, J. R. Soc. Interface. 10, 20130361, http://doi.org/10.1098/rsif.2013.0361

Minor comments.

- Line 33: “The butterfly has two characteristics, two pairs of broad wings and low flapping frequency” Add i.e. between characteristics and two

- Eq. (2) and elsewhere: Not all variables have been defined.

- Line 148: N = 4. Does this relate to the number of different butterfly specimens or the number of measurements on the same butterfly specimen? Please clarify.

- Line 150: thickness is 2.5% of the mean chord. Please specify where this number comes from.

- Grid sensitivity: difficult to follow how the resolution was increased in 3D.

Please check the manuscript for grammar.

Author Response

(The authors gave the same response as above.)

Reviewer 3 Report

The paper uses experimental and computational methods to evaluate the effects of chordwise wing flexibility on butterfly flight.

These are some of my comments:

1. The experimental measurement of the wing kinematics is not used as inputs while prescribing the wing motion in the numerical simulations. This would have helped in ensuing better comparison of the PIV and the CFD flow field. For eg. did the stroke plane angle vary between up and downstrokes? Was the deformation assumed to be symmetric during up and down strokes? 

Instead the authors seem to be using measurement from a previous study [22]?

2. A related concern is that the three dimensional wing deformation is prescribed to the CFD model rather than using coupled FSI solver. The prescribed deformation is also not informed from the deformation from the experimental measurements. Without a comparison between prescribed and measured wing deformation, the flow field from the prescribed CFD simulation (and the resulting lift and thrust measurements) will not be representative of actual flight conditions. 

3. The authors seem to have not discussed their current results in comparison to some of the previously published key papers in regards to flow field around freely flying butterflies. For eg. "Unconventional lift-generating mechanisms in freely-flying butterflies" - Srygley and Thomas, Nature, 2022 

The quality of English language is poor at several places in the manuscript and requires significant revision and grammatical checks. 

Author Response

(The authors gave the same response as above.)

Round 2

Reviewer 2 Report

I thank the authors for responding to my comments. However, I am of the opinion that my concerns have not been adequately addressed. 

A. Reproducibility. The methods and results should be documented in such a way that these could be reproduced. However, as noted in my original review (#2, #4), there are certain important parameters that don’t seem to be reported in a precise way:

* For example, the wing geometry and reference axes in the schematic shown in Fig. 4a is not the same as the ones used in the study (Fig. 5), especially at the wing root.

* Furthermore, the camber deformation measurements seem to be the highlight of the study. Yet, there is almost no report of the measurement data, except for Fig. 4b. Instead, authors should carefully describe the measurement results including the values of a(t), b(t), and l1, l2 and how the camber varied wrt the span in the measurements. 

* Lines 245-246: “In the simulation, the wing model was controlled by the same characteristic camber as the chordwise plane at the wing root position, and it was reconstructed as a 3D wing.” This sentence seems to suggest that the same camberline (Eq. 3) was applied at all wing spanwise locations. If so, this should be made clear for example by saying that Eq. 3 was applied at all z_w locations resulting in a spanwise uniform camber.

The above point is particularly important because lines 217-218 say that “the camber deformation is dependent on the chordwise direction, the value of which will vary with the spanwise position and time,” which is the case with real butterfly wings. But in the simulations this was simplified.  

* Lines 196 - 197: “These data were then used to prescribe the wing flapping motion and body angle in the simulation.” Were any pitching/feather and deviation angles measured in the experimental observations? The butterfly wing motion is highly three-dimensional. However, it is unclear how the pitching and deviation angles varied in the wing measurements and if these were prescribed in the simulations.

B. Discussion of the limitations of the methods (#5). The revised version of the manuscript still lacks a discussion of how the limitations of the employed methods (e.g. 1-way FSI in place of 2-way FSI, uniform camber in place of spanwise varying camber, etc.) affect the main results and conclusions of this study.

C. Body angle (#1). In the revised abstract, the authors state that “The wing undergoes a significant chordwise deformation which can generate a larger lift coefficient as that with a higher body angle, resulting in a 14% increase compared to lower chordwise deformation and body angle.” 

This seems to suggest that a higher body angle results in a higher lift coefficient. I have two major concerns with this:

i) Isn’t the lift coefficient also a function of the wing kinematics: flapping angle, pitching angle, deviation angle in a relatively minor way, and stroke plane angle? It seems to be very simplistic to say that a higher body angle produces a higher lift coefficient.

ii) Isn’t the body angle of a butterfly also an outcome of the butterfly dynamics? Because of the relatively large wings, the body rotates significantly during flight as also shown in Fig. 3e. I am of the opinion that the butterfly doesn’t “control” its body motion, but the observed body rotation is closely coupled with the wing motion and the resulting aerodynamics.

D. PIV vs CFD (#4). The authors say in the response that “in our PIV experiment, the flow field was captured in a stationary condition from a forward and slightly downward angle while the butterfly was flying slowly forward. Due to the recording angle, we observed the downwash airflow and vortices generated during the upstroke and downstroke of the butterfly's wings, as shown in Figure 7. In our simulation results, as shown in Figure 10, it was observed that the position of the detached vortex during the downstroke and upstroke was the same compared to the PIV experiment.”

One of the strengths of this study is the use of the combined CFD and PIV approach. The CFD results should be placed in Fig. 7 next to the experimental results so that the readers can compare these side-by-side. Any differences should be noted and discussed. 

E. Other minor comments:

* Line 159: “The thickness of the wing model was 2.5% of the mean wing chord, which was exactly the right size for the grids to fill up” 

Did the authors measure the thickness of the butterfly wings? The ease of grid generation is not a good reason in a scientific study in general.

* Lines 180-181: “The three different grid sizes were adjusted by reducing the grid size to increase the resolution of the computational domain.”

Can the authors be more specific for example by reporting the resolution of the grids on the wing and the first grid spacing away from the wing? It is important for the readers to be able to follow the near wing grid resolution especially if there is almost no grid sensitivity (Fig. 3c).

* Abstract: “The results show that chordwise deformation enhances wing performance due to the increase of the strength of vortex and more stabilized attached vortex.”

It is well-known that for example a negatively cambered airfoil could reduce the lift production. Also, if the wing is too flexible (e.g. flag / paper) the resulting vortex dynamics may not positively affect the aerodynamics. As such this sentence needs to be qualified by stating under which conditions or by which mechanism positive effects may arise.

F. Comment not raised in my original review. I apologize that I missed this in my original review but I think this is important to mention.

* Lines 350 - 351: “The curvature of the leading edge of the wing diverted the horizontal force to the vertical direction during the upstroke, resulting in the jet flow being directed downward.” The following paper pointed out a similar FSI mechanism that the chordwise flexibility can redistribute lift versus thrust by changing the projection angle of the wing with respect to the main flow  by changing airfoil via camber deformation. As such, this paper should be referenced here.

W. Shyy, H. Aono, S.K. Chimakurthi, P. Trizila, C.-K. Kang, C.E.S. Cesnik, H. Liu, Recent progress in flapping wing aerodynamics and aeroelasticity, Progress in Aerospace Sciences,

Volume 46, Issue 7, 2010, Pages 284-327

The manuscript should be carefully proofread to correct the many grammar mistakes and improve the quality of the scientific writing. 

Author Response

Thank you very much for the valuable comments provided by the editor, Gu, and the reviewers in the second round. We have included our organized answers and point-by-point responses to the reviewers in the attached file. For your consideration, we have highlighted the revised parts in the manuscript. Additionally, we have enlisted the assistance of an English editor from MDPI to revise the entire paper, ensuring smooth readability for the readers.

Reviewer 3 Report

My previous comments were addressed satisfactorily by the authors.

However, I do have several minor corrections and questions for the authors:

  1. For body angle magnitude, the manuscript reports both single decimal precision as well as two decimal precision at several places in the text and figures (39.41 vs 39.4 deg, 55.01 vs 55.0 vs 55 deg, 75.41 vs 75.4 deg). This needs to be unified.

  2. Units of velocity: both m/s as well as m s-1 are used. Please unify.

  3. Table 2 caption: “wing aerodynamics” instead of “wing aerodynamic”

  4. In Figure 6(a) and (b) it is better to enclose the y-axis label units in braces to avoid any confusion

  5. Is it possible to include the non-dimensional time in Figure 7 at the three time instants shown?

  6. In Figures 9, 10, 12 the color bar labels and units should be next to each other for clarity rather than on the either ends of the color bar (similar to Figure 7)

  7. Line 265/266: “Each group of the recorded video during the butterfly flight generally contains 80 to 110 data”. 

    Please replace ‘data’ with ‘measurements’ or ‘observations’

  8. Line 328: include space between ratio and the bracket.

  9. In paragraph between lines 363 and 377, it is better to change LEVu to LEVu (subscript) for consistency with labels in Figure 9 and Figure 10.

  10. Line 393 and 394: “The variations of total force and power at both angles were relatively minor and reach approximately -18% and 12%”. Please verify these percentages compared to the values stated in Table 3.

Minor editing required.

Author Response

(The authors gave the same response as above.)
